# Effect of Chitosan on Rheological, Mechanical, and Adhesive Properties of Pectin–Calcium Gel

**DOI:** 10.3390/md21070375

**Published:** 2023-06-25

**Authors:** Sergey Popov, Nikita Paderin, Elizaveta Chistiakova, Dmitry Ptashkin, Fedor Vityazev, Pavel A. Markov, Kirill S. Erokhin

**Affiliations:** 1Institute of Physiology of Federal Research Centre “Komi Science Centre of the Urals Branch of the Russian Academy of Sciencesk”, 167982 Syktyvkar, Russia; paderin_nm@mail.ru (N.P.); kvashninova.e@yandex.ru (E.C.); ptdimas@ya.ru (D.P.); rodefex@mail.ru (F.V.); 2The Federal State Budgetary Institution “National Medical Research Center of Rehabilitation and Balneologyk”, 121099 Moscow, Russia; p.a.markov@mail.ru; 3N.D. Zelinsky Institute of Organic Chemistry, Russian Academy of Sciences, 119991 Moscow, Russia; erokhin@ioc.ac.ru

**Keywords:** chitosan, pectin, ionotropic gel, rheology, mechanical properties, weight loss, serosal adhesion

## Abstract

In the present study, chitosan was included in the pectin ionotropic gel to improve its mechanical and bioadhesive properties. Pectin–chitosan gels P–Ch0, P–Ch1, P–Ch2, and P–Ch3 of chitosan weight fractions of 0.00, 0.25, 0.50, and 0.75 were prepared and characterized by dynamic rheological tests, penetration tests, and serosal adhesion ex vivo assays. The storage modulus (G′) and loss modulus (G″) values, gel hardness, and elasticity of P–Ch1 were significantly higher than those of P–Ch0 gel. However, a further increase in the content of chitosan in the gel significantly reduced these parameters. The inclusion of chitosan into the pectin gel led to a decrease in weight and an increase in hardness during incubation in Hanks’ solution at pH 5.0, 7.4, and 8.0. The adhesion of P–Ch1 and P–Ch2 to rat intestinal serosa ex vivo was 1.3 and 1.7 times stronger, whereas that of P–Ch3 was similar to that of a P–Ch0 gel. Pre-incubation in Hanks’ solution at pH 5.0 and 7.4 reduced the adhesivity of gels; however, the adhesivity of P–Ch1 and P–Ch2 exceeded that of P–Ch0 and P–Ch3. Thus, serosal adhesion combined with higher mechanical stability in a wide pH range appeared to be advantages of the inclusion of chitosan into pectin gel.

## 1. Introduction

Surgical removal of a fragment of the intestine is often performed in the treatment of gastrointestinal cancer, bowel obstruction, diverticulitis, and Crohn’s disease. Anastomosis, e.g., an artificial connection of the two remaining ends of the bowel, must be established following procedures involving intestinal resection [1]. Anastomotic leakage, which is documented in 1–19% of patients, represents one of the harmful complications that greatly increases the risk of postoperative mortality and morbidity [2]. To treat anastomotic defects, various surgical approaches have been developed in the last few decades, including stenting, reinforcing strips, buttressing, and tissue adhesives [1,3]. In clinical studies, commercially available adhesives such as PEG-based materials, cyanoacrylate, and fibrin glue have been utilized to seal and mechanically support the anastomosis [4]. Their utilization is nevertheless constrained by issues including poor underwater adhesive strength, acidic breakdown, low wound healing capacity, and risks of cytotoxicity. In recent years, hydrogel adhesives have been investigated for supporting the anastomotic site because of their atraumatic nature, fluid-tight sealing capability, favorable wound healing properties, and facile application [5].

Hydrogels are three-dimensional, cross-linked networks of polymer chains capable of absorbing and retaining significant volumes of water in the interstitial spaces between chains. Hydrogels are commonly utilized in applications that interact intimately with biological organisms, such as tissue engineering, drug delivery, and biological research [6,7,8]. Softness, wetness, responsiveness, and biocompatibility contribute to their intensive research as machines, sensors, actuators, coatings, etc. [9,10,11]. Natural polysaccharides for hydrogel preparation have advantages such as bioactivity, low toxicity, biocompatibility, and similarity of structure and function to the glycans of extracellular matrix [12,13]. Pectin is a gelling heteropolysaccharide isolated from higher plants that is widely used in several scientific fields [14,15]. Pectins include a diverse group of polyanionic polysaccharides whose backbone is formed by 1,4-linked α-D-galacturonic acid (GalA) residues that can be partly methyl esterified. Pectins with a low degree of methyl esterification form a gel in the presence of multivalent cations in a wide pH range, whereas high-methyl-esterified pectins form physical gels at pH < 3.5 in the presence of co-solutes [16]. Biodegradability, biocompatibility, drug loading/releasing capacity, and tunable mechanical properties contribute to the application of pectin gels as promising materials for biomedicine [17]. Pectin has been found to attach to the glycocalyx of visceral mesothelium [18] and was proposed as a serosal bioadhesive for pleural air leaks [19,20,21] and a sealant for anastomotic healing after intestinal surgery [22]. Recently, our group showed that a hydrogel made of ionically cross-linked apple pectin adhered to rat intestinal serosa ex vivo with a strength that depended on the type of cross-linking cation [23]. However, the serosal adhesion and strength of the ionotropic pectin gel were found to decrease significantly after incubation in physiological media [23,24,25]. Improvement of adhesivity and strengthening of the pectin gel network appear to be necessary to obtain promising hydrogel adhesives to treat anastomotic leakage.

Chitosan is a natural polymer derived by the partial deacetylation in an alkaline solution of chitin, which is the main component of the exoskeleton of crabs, shrimp, and krill. It is a linear amino-polysaccharide of β (1-4)-linked D-glucosamine and N-acetyl-D-glucosamine residues and has similarity to glycosaminoglycans of the extracellular matrix [26]. Chitosan is used as a hydrogel in many successful applications within the biomedical field, including wound dressing, drug controlled-release systems, tissue engineering, and regenerative medicine [26,27,28]. The use of chitosan as a bioadhesive substance is one area of considerable attention. The ability to adhere to the skin and mucous membranes are the advantages of chitosan for the development of dressing materials and targeted drug delivery, respectively [29,30,31,32]. The cationic behavior of chitosan in acidic solutions and the induction of gelation reactions mediate interactions with other materials and the formation of composites. In particular, chitosan, due to the presence of primary amino groups, can produce polyelectrolyte complexes with anionic polymers such as pectin [33]. Pectin/chitosan systems with improved properties compared to single-component systems have already been proposed for 3D bioprinting [34,35], nanodressing [36], scaffolds for bone regeneration [37], buccal [38] and vascular [39] patches, pH sensors [40], etc. In this paper, chitosan has been selected to improve mechanical properties and enhance serosal adhesion of pectin gel to serosa. Chitosan has been previously shown to alter the rheological properties of concentrated pectin solutions by acting as a crosslinker [41]. Single systems containing only pectin or chitosan have been found to present a predominantly liquid-like behavior with an extremely low viscosity consistency index value [42]. Binary systems containing both pectin and chitosan show a typical gel-like behavior where the value of the storage modulus (G′) is greater than the loss modulus (G″) value [34]. However, the tan δ value (loss angle) of pectin–chitosan hydrogel has been found to be greater than 0.1, indicating that pectin–chitosan hydrogels are rather weak gels [34]. Therefore, the rheological properties of the pectin–chitosan complex are well suited for 3D printing [34,35,43,44] and injectable systems [33,45] where flow behavior is required. The use of calcium ions as an additional cross-linker was proposed in the present study to improve the gel properties of the pectin–chitosan complex and achieve the mechanical strength required for a serosal patch. An increase in the viscoelasticity of pectin–chitosan hydrogel in the presence of Ca^2+^ ions has been previously shown in a number of investigations [39,46,47,48]. We hypothesized here that the crosslinking intensity in the pectin-calcium-chitosan system would be influenced by its rheological properties.

The aim of this research was to investigate the effect of chitosan on the strength and serosal adherence of an ionotropic pectin gel. The rheological properties of pectin–chitosan gels of various concentrations, as well as fibroblast adherence to them, have also been elucidated.

## 2. Results

### 2.1. Mechanical and Rheological Properties of Pectin–Chitosan Gels

Pectin–chitosan gels were prepared by immersing a dialysis bag containing a heated mixture solution of pectin and chitosan (80 mL) in a calcium chloride solution (250 mL). After 24 h, a solid gel was formed in the dialysis bag in the form of a cylinder with a length of 10–11 cm and a diameter of 2.4–2.8 cm. The cylindrical gel was then cut into puck-shaped samples at a predetermined height for subsequent analyses.

Pectin–chitosan gels P–Ch0, P–Ch1, P–Ch2, and P–Ch3 with chitosan weight fractions with respect to the total polymer content of 0.00, 0.25, 0.50, and 0.75 were obtained using pectin:chitosan (g:g) ratios of 4:0, 3:1, 2:2, and 1:3, respectively. However, a 4% solution of chitosan (pectin:chitosan 0:4) did not form a gel under these conditions.

The inclusion of chitosan into the pectin gel did not affect its density, but it increased the water content and decreased the pH (Table 1). The calcium content in P–Ch0 and P–Ch1 gels did not differ significantly and was 12.6 ± 1.9 and 8.9 ± 2.4 mM/g, respectively (*p* > 0.05, *n* = 8). The P–Ch2 and P–Ch3 contained 6.9 ± 1.9 and 6.3 ± 1.9 mM of calcium per g of gel, respectively, which was 1.8 and 2 times less than in the P–Ch0 gel (both *p* < 0.05, *n* = 8). The hardness, elasticity, and Young’s modulus of P–Ch1 pectin gel were 2.9, 1.7, and 1.5 times higher than those of P–Ch0 gel. The hardness, elasticity, and Young’s modulus of P–Ch2 gel were also higher than those of P–Ch0 gel but significantly lower than those of P–Ch1 gel. The hardness and YounG′s modulus of P–Ch3 gel were lower than those of the chitosan-free pectin gel P–Ch0 (Table 1).

Small-deformation dynamic rheological measurements were conducted by applying small oscillating stresses or strains and recording the responses of the gels. Strain sweep experiments were performed from 0.01 to 100% strain amplitude using a controlled shear rate mode at 20 °C at a constant frequency and stress of 1 Hz and 9.0 Pa, respectively. The data were analyzed in the linear (LVE) and nonlinear (n-LVE) viscoelastic regions. The storage modulus G′ of all the gel samples was greater than the loss modulus G″ throughout the LVE region (Figure 1).

The G′_LVE_ and G″_LVE_ values of P–Ch1 gel were 1.9 and 2.1 times higher than those of P–Ch0 gel, respectively (Table 2). However, a further increase in the content of chitosan in the gel significantly reduced the G′_LVE_ and G″_LVE_ values. The G′_LVE_ and G″_LVE_ values of P–Ch3 gel were 6.9 and 9.5 times lower than those of chitosan-free pectin P–Ch0 gel (Table 2). The loss factor, tan [δ]_LVE_ was 0.14–0.21 for P–Ch0–P–Ch3, indicating a solid-like behavior of pectin–chitosan gels. Changes in the limiting value of stress (τL), fracture stress (τFr), and corresponding complex modulus (G*_FP_) with the stress at the flow point, caused by the addition of chitosan, occurred similarly to changes in the G′_LVE_ and G″_LVE_ values: values for P–Ch1 were maximum and for P–Ch3 were minimum among pectin–chitosan gels (Table 2). The fracture strain (γFr) measured at the fracture points of P–Ch2 and P–Ch3 was higher than in P–Ch0 and P–Ch1. Chitosan did not affect the slope of the loss tangent in the nonlinear region (tan [δ]_AF_).

The rheological parameters of the frequency sweep tests in the LVE range are shown in Table 3. G′ and G″ frequency dependencies are described by the power law equation with high correlation (*R*^2^ > 0.99). At all frequencies, P–Ch1 gel showed higher values, whereas P–Ch2 and P–Ch3 gels had lower G′ values than P–Ch0 gel. The frequency dependences of the elastic (k′), loss (k″), and complex (A) moduli increased in P–Ch1 compared to chitosan-free pectin gel P–Ch0. However, the values of these parameters, as well as the complex viscosity slope (η × s), decreased significantly with a further increase in the content of chitosan. The overall loss tangent (k″/k′) did not differ among pectin–chitosan gels. The gels seemed to show elastic behavior, as revealed by the low slope of both moduli, indicated by 0.08 ≤ n′ ≤ 0.10 and 0.01 ≤ n″ ≤ 0.05 (Table 3).

The viscosity values of P–Ch1 were the highest and the lowest for P–Ch3 among pectin–chitosan gels (Table 4).

Scanning electron microscopy (SEM) showed that freeze-dried samples of pectin–chitosan gels have a loose, porous internal structure (Figure 2A–D). Although exact pore sizes have not been measured, micrographs with a magnification of 50× illustrate that the shape and size of the pores of the gel matrix changed with the increasing content of chitosan in the gel. The pore size of pectin gel P–Ch0 can be estimated to be approx. 150–200 µm, while the pores of the P–Ch1 and P–Ch2 gels were more elongated and much larger (ca. 400–600 µm). The P–Ch3 gel had rounded pores with a size close to that of the P–Ch0 gel (Figure 2A–D). Surface images taken at higher magnification (2000×) showed that the P–Ch2 gel had the highest surface roughness (Figure 2E–H).

EDS analysis revealed that the calcium content in P–Ch0 gel was equal to 17 ± 3.6 wt%. P–Ch1, P–Ch2, and P–Ch3 gels contained 13 ± 3, 11 ± 1, and 12 ± 2 wt% of calcium, i.e., 24, 35, and 29% lower than P–Ch0 gel (all *p* < 0.05, *n* = 5). Pectin–chitosan gels were similar in their contents of carbon (27–34 wt%), oxygen (22–30 wt%), and chlorine (23–31 wt%).

### 2.2. Stability of Pectin–Chitosan Gels Incubated at Different pH

The weight and mechanical properties were measured during the incubation of P–Ch0, P–Ch1, P–Ch2, and P–Ch3 in Hanks’ solution at pH 7.4, 5.0, and 8.0 for 24 h. The weight of chitosan-free pectin gel P–Ch0 did not change during incubation in Hanks’ solution at any pH (Figure 3). The inclusion of chitosan in the pectin gel led to a decrease in the weight of the samples during incubation. The weight of gel P–Ch1 decreased by 14, 22, and 12% of the initial value after 24 h of incubation at pH 5.0, 7.4, and 8.0. The weights of gels P–Ch2 and P–Ch3 decreased by 18, 31, and 49% and 10, 38, and 60%, respectively, at pH 5.0, 7.4, and 8.0 (Figure 3).

Galacturonic acid (GalA) was found to release from the gels incubated in Hank’s solution (Figure 4). The concentration of GalA in the incubation medium of P–Ch0 and P–Ch1 gels was significantly higher than that of P–Ch2 and P–Ch3 gels during incubation at pH 5.0. The release of GalA from the P–Ch0 gel did not depend on the pH of the incubation medium; the GalA concentration was about 500 µg/mL after 24 h of incubation. The release of GalA from the P–Ch1, P–Ch2, and P–Ch3 gels increased significantly with the increasing pH of the incubation medium.

Figure 5 shows the effect of pectin–chitosan gels on the pH level of Hank’s solution during incubation. The P–Ch0 and P–Ch1 gels lowered the pH of the incubation medium from 5.0 to 3.9–4.0, from 7.4 to 6.6–6.9, and from 8.0 to 7.2–7.5. The P–Ch2 gel reduced pH to a lesser extent than P–Ch0 and P–Ch1, whereas the gel P–Ch3 did not affect the pH level of the solution.

The mechanical properties of P–Ch0 gel gradually decreased upon incubation in Hanks’ solution at pH 5.0, 7.4, and 8.0 (Figure 6, Figure 7 and Figure 8). Hardness, elasticity, and YounG′s modulus of P–Ch0 gel were 20–29, 68–69, and 21–30% of baseline after 24 h of incubation at pH 5.0, 7.4, and 8.0 (Table 5). P–Ch1 gel lost its mechanical properties during incubation to a lesser extent than P–Ch0 gel. The hardness, elasticity, and Young’s modulus of P–Ch1 gel were 52, 73, and 91% of baseline after 24 h of incubation at pH 5.0 (Table 5). The hardness of P–Ch2 and P–Ch3 gels did not change upon incubation at pH 5.0 and increased upon incubation at pH 7.4 and 8.0 (Figure 6). The elasticity of P–Ch2 and P–Ch3 gels increased relative to the initial level upon incubation at all pHs studied (Figure 7).

### 2.3. Serosal Adhesion of Pectin–Chitosan Gels

Chitosan was found to increase the serosal adhesivity of the pectin gel. The adhesion strength of the P–Ch1 and P–Ch2 gels to the rat serosa was 25 and 71% higher than that of the chitosan-free pectin gel P–Ch0 (Figure 9A). However, the adhesion of P–Ch3 to the serosa did not differ from that of P–Ch0.

The adhesion strength of P–Ch0 gel decreased from 75.7 ± 19.7 to 49.2 ± 17.1 and 47.0 ± 9.5 mN after 24 h of incubation in Hank’s solution of pH 5.0 and 7.4, respectively (Figure 9B,C). Incubation in Hank’s solution at pH 8.0 did not affect the adhesiveness of P–Ch0 (Figure 9D). The adhesion strength of the P–Ch1 remained at ca. 100 mN after incubation at pH 5.0, 7.4, and 8.0, which exceeded the adhesion strength of the P–Ch0 gel by 1.7–2.2 times (Figure 9B–D). The adhesion strength of the P–Ch2 gel decreased after incubation at pH 5.0 and 7.4 and did not change after incubation at pH 8.0, remaining higher than that of P–Ch0. Adhesion of P–Ch3 gel to serosa decreased by 37 and 43% after 24 h of incubation in Hank’s solution at pH 5.0 and 7.4, respectively, and did not change after incubation at pH 8.0 (Figure 9B–D).

The work of adhesion of the P–Ch1 and P–Ch2 gels before incubation in Hank’s solution did not differ from that of the P–Ch0 gel despite the higher adhesion strength (Figure 10A). The work of adhesion of P–Ch1 and P–Ch2 gels incubated in Hanks’ solution at pH 5.0, 7.4, and 8.0 exceeded that of the P–Ch0 gel, accompanying an increase in adhesion strength (Figure 10B–D). Interesting results were obtained with respect to P–Ch3 gel. Although the adhesion force of the P–Ch3 gel to the serosa did not differ from that of the P–Ch0 gel, the work of adhesion of the P–Ch3 gel was higher than that of the P–Ch0 gel. In particular, the work of adhesion of the P–Ch3 gel after incubation in Hanks’ solution with pH 5.0 and 7.4 was 1.6 and 1.7 times higher, respectively, than that of the P–Ch0 gel (Figure 10B,C).

Figure 11 shows representative adhesion profiles of P–Ch0 and P–Ch3 gels to illustrate the observed pattern. The adhesion profiles included the deadhesion phase (Figure 11A(i)) with a peak force corresponding to the adhesion strength and a debonding curve (Figure 11A(ii)), during which the adhesion force decreased because of the gradual breaking of bonds between the gel surface and the serosa. As can be seen, the width of the P–Ch3 profile was larger than that of the P–Ch0, resulting in a higher work of adhesion, which was determined as the area under the force overtime curve.

### 2.4. Adhesion of Fibroblasts to Pectin–Chitosan Gels

Human fibroblasts weakly adhered to the surface of pectin–chitosan gels. The number of adhered fibroblasts on the culture plastic surface (control) was 52.4 ± 9.2 cells/0.1 mm^2^ after 2 h of incubation. The cells were rounded, with an average diameter of approx. 25 µm (Figure 12A). Subsequently, the number of adherent fibroblasts did not change significantly; however, more than 90% of the cells spread out on the plastic surface and acquired a spindle shape after 24 h of incubation (Figure 12B; Table 6). After 2 h of incubation, fibroblasts on the surface of pectin–chitosan gels had a rounded shape, but their number was 2.2–3.4 times less than on the plastic surface. The number of adherent fibroblasts on the surface of the chitosan-free gel P–Ch0 after 24 h decreased by 3.7 times compared with 2 h of incubation and amounted to only 9% of the number of cells adhered to the plastic (Table 6). An increase in the content of chitosan in the gel led to an increase in the number of fibroblasts that adhered within 24 h. The number of fibroblasts adhered to the P–Ch1, P–Ch2, and P–Ch3 gels was 2.2, 3.6, and 4.6 times higher than on the P–Ch0 gel (Table 6). It should be noted that cells on the surface of pectin–chitosan gels retained their round shape even after 24 h of incubation (Figure 13).

## 3. Discussion

The main finding of the present study is that chitosan was shown to improve the tissue adhesiveness of pectin gel cross-linked with calcium ions. Although the high tissue adhesiveness of chitosan gels was shown in relation to the skin, the adhesion of chitosan gels to the serous surface has not been previously studied. In particular, in the study [31], the adhesion strength of chitosan gels to porcine skin was approx. 7 kPa. Considering a contact area of 81 mm^2^, the serosal adhesion strengths of the pectin–chitosan gels were in the range of 76–130 mN, which corresponds to 0.9–1.6 kPa. The higher skin adhesion values may be due to the higher compressing force of the gel sample on the porcine skin at the start of the assay than on the more delicate intestinal serosa of the rat in the present study (60 and 1.2 kPa, respectively). The adhesion strength of the chitosan gel to the skin of a newborn rat was 0.12–0.18 kPa after compressing the sample with a force of 0.04 kPa [29]. Previously, it was also shown that the adhesion strength of pectin hydrogel cross-linked with Fe^3+^ ions is 6.4 kPa [23]. However, the tissue adhesion assay, in particular the preparation of the gel specimen, was modified in the present study compared to the method used in the previous study, which may explain the difference in results. In addition, Fe^3+^ pectin gel was shown to demonstrate less stability upon implantation than Ca^2+^ gel, cause a pro-inflammatory response, and decrease the viability of human peripheral blood mononuclear cells in vitro [23,24]. Therefore, calcium ions were chosen as a cross-linker to obtain pectin gel in the present study. The adhesion to the serosa of the rat small intestine of the fresh P–Ch1 and P–Ch2 gels containing chitosan weight fractions of 0.25 and 0.5, respectively, was 1.3 and 1.7 times stronger than that of the chitosan-free gel P–Ch0. An unexpected finding was that increasing the content of chitosan to 0.75 of chitosan weight fractions did not lead to a further increase in adhesion strength but, on the contrary, reduced it to the level of a chitosan-free pectin gel.

It was revealed that pectin–chitosan gels with different tissue adhesiveness differ in rheological and structural-mechanical properties. According to [49], the rheological characteristics of hydrocolloids may be classified into characteristics related to the strength, number, distance of links, and timescale of the junction zone. The increase in such rheological parameters as G′_LVE_, τL, G*_FP_, A, k′, and k″ clearly indicated an increase in the strength of the linkage in the P–Ch1 gel compared to the P–Ch0 gel. These changes were consistent with the increase in hardness, elasticity, and Young’s modulus of the P–Ch1 gel. Such parameters as τFr, n′, and E indicate an increase in the number of linkages in the P–Ch1 gel compared to the P–Ch0 gel and a decrease in the P–Ch2 and P–Ch3 gels compared to the P–Ch1 gel. The addition of chitosan did not affect the rheological parameters associated with the distance of links (n″) and the timescale of the junction zone (Tan [δ]_LVE_, k″/k′, γFr).

In general, dual network hydrogels or interpenetrating polymer network hydrogels can be created by adding a second material to the initial hydrogel [33]. The formation of pectin–chitosan gel in our work presumably occurred due to both the cross-linking of pectin chains with calcium ions and the formation of a polyelectrolyte complex between pectin and chitosan molecules (Figure 14). Low-methyl esterified pectin forms hydrogel in the presence of calcium ions by the so-called “egg boxk” mechanism, which proposes two ionic linkages between the free carboxylic acid groups of two adjacent pectin chains (Figure 14A). Egg boxes are also supported by hydrogen bonds and van der Waals interactions in addition to ionic bridges [16,50]. In chitosan-containing gels, an ionic bond between the amino groups of chitosan and the carboxylic groups of pectin is formed, leading to the formation of a polyelectrolyte complex as depicted in Figure 14C,F. The number of cross-links of two -COO^−^ groups through the calcium ion and cross-links of -COO^−^ groups with -NH3^+^ groups probably depends on the content of pectin and chitosan in the gel. The theoretical amount of -COO^−^ and -NH_3_^+^ groups in the gels may be calculated taking into account the concentration of pectin and chitosan and their degree of methyl esterification and deacylation, respectively, according to the formulas:ν(-COO^−^) = (m(p)/M(p)) × (1 − DM)(1)
ν(-NH_3_^+^) = (m(ch)/M(ch)) × (DD)(2)
where ν(-COO^−^) is the amount of substance with free carboxyl groups; m(p) is the weight of dissolved pectin; M(p) is the molecular weight of galacturonic acid; DM is the degree of methyl-esterification; ν(-NH_3_^+^) is the amount of substance with free amine groups; m(ch) is the weight of dissolved chitosan; M(ch) is the molecular weight of D-glucosamine; DD is the degree of deacetylation.

According to the calculations, the P–Ch0 gel contains 123 mM -COO^−^ groups available for the formation of egg-box cross-links. There are 93 and 50 mM -COO^−^ and -NH_3_^+^ groups in the P–Ch1 gel, respectively, indicating that only 43 mM -COO^−^ groups may be available for egg-box cross-linking. Fifty out of the 93 mM free -COO^−^ groups of pectin are most likely involved in cross-linking with the -NH_3_^+^ groups of chitosan, due to the fact that the cooling of the polymer mixture during the formation of the gel occurs faster than the diffusion of calcium ions into the gel body. In the P–Ch1 gel, ionic bonds between amino and carboxylic groups reinforce the cross-linking of pectin molecules by calcium ions, which leads to the strengthening of the gel network (Figure 14B).

The total number of cross-links decreases with a further increase in the chitosan content in P–Ch2 gel, while the number of positively charged amide groups exceeds the number of carboxyl groups due to a decrease in the pectin content (Figure 14D). According to the calculations, the P–Ch2 gel contains 62 and 100 mM -COO^−^ and -NH_3_^+^ groups, respectively. The decrease in the theoretically calculated number of potential cross-links in the P–Ch2 gel compared to the P–Ch1 gel is confirmed by a decrease in gel hardness and a decrease in the rheological characteristics associated with the number of linkages.

The number of -NH_3_^+^ groups even more significantly exceeds the number of -COO^−^ groups in the P–Ch3 gel (31 vs. 150 mM). The lack of carboxyl groups in the P–Ch3 gel reduces the possibility of the formation of ionic bonds, explaining a weakening of the gel network. In addition, an excess of -NH_3_^+^ groups due to the high content of chitosan can probably be associated with the repulsion of a large number of unbound positively charged amide groups, which make the gel network very weak (Figure 14D). A similar pattern was found by Jindal et al. [51], who showed that the strength of the pectin–chitosan film first increased and then decreased as the chitosan content increased above a certain optimal value.

The data obtained suggest that the gels obtained, depending on their composition, can be assigned to three types of gel networks. In P–Ch0 gel, the network is formed only by the calcium egg box, which links two pectin chains. The network in P–Ch1 gel seems to be created by the calcium ion cross-linking of pectins as well as the cross-linking of pectin and chitosan molecules via a -COO^−^ and -NH_3_^+^ linkage. In P–Ch2 and P–Ch3 gels, the gel network is predominantly formed by crosslinking pectin and chitosan molecules through -COO^−^ and -NH_3_^+^ bonds.

The proposed scheme makes it possible to explain the differences in the calcium content of the gels. The highest content of calcium in the P–Ch0 gel was presumably due to the involvement of calcium in the egg-box structure. P–Ch1, P–Ch2, and P–Ch3 gels with low or no calcium cross-links were found to have significantly less calcium than P–Ch0 gel. Although chitosan has little adsorption in relation to Ca^2+^, the latter can be complexed with –NH_2_, CH_3_CONH, and –OH groups of chitosan [52]. Therefore, the presence of calcium in gels containing chitosan may be due to the possible formation of Ca^2+^-chitosan complexes. Changes in the mechanical and rheological properties of the gel upon the addition of chitosan are accompanied by a change in the internal structure of the gel, as revealed by SEM.

Functional stability in physiological solutions is an important characteristic of biomaterials. Here, it is shown that the wet pectin gel did not change its weight upon incubation in Hanks’ solution with pH 5.0, 7.4, and 8.0. The data obtained indicate that the pectin gel was in an equilibrium state between the osmotic and elastic parts of the free energy balance. Gels containing chitosan lost weight during incubation, probably due to dehydration. Neufeld and Bianco-Peled [53] demonstrated that the deionization of the -NH_3_^+^ groups of chitosan can account for the shrinkage of the gel network and the displacement of water from the pectin–chitosan gel, the so-called de-swelling phenomenon. Our results are consistent with this view, as weight loss during incubation of pectin–chitosan gels increased with increasing pH of the incubation solution. The chitosan becomes more uncharged and less soluble as the pH increases, resulting in a decrease in the electrostatic repulsion of the molecules, so that the gel becomes more compact. The presence of calcium cross-links in P–Ch1 gel apparently makes its network more resistant to reduction of the gel mesh size and dehydration.

Chitosan was found to affect the stability of the gel network at different pH levels, as measured by GalA release. P–Ch0 gel, formed only by calcium cross-links, released an equal amount of GalA at pH 5.0, 7.4, and 8.0. Anions contained in Hanks’ solution, such as H_2_PO_4−_ and HPO_4_^2−^, were supposed to bind calcium ions, causing the “egg-boxk” structures to disintegrate. As noted above, the -NH_3_^+^ groups of chitosan cross-linked with the -COO^−^ groups of pectin GalA play a key role in the formation of the gel network in P–Ch1, P–Ch2, and P–Ch3 gels. Therefore, it is likely that the transformation of -NH_3_^+^ into NH_2_ groups with increasing pH contributed to the release of GalA.

The change in the concentration of GalA can explain the observed change in the pH of the incubation medium. The greatest release of GalA from P–Ch0 and P–Ch1 gels was accompanied by the greatest acidification of the medium during incubation at pH 5.0 and 7.4. The highest release of GalA from P–Ch1 gel corresponded to the highest acidification of the medium during incubation at pH 8.0. P–Ch3 gel had little effect on the pH of the incubation solution, probably due to the low content of pectin in its composition. It is well known that pH value impacts enzyme activity, fibroblast activity, and immunological responses, making it one of the crucial elements in the healing process of both acute and chronic wounds [54]. The alkaline milieu has been shown to promote a lower rate of healing, while the acidic milieu is essentially a physiological barrier to bacterial growth and prevents infection [55]. Therefore, acidification of the medium in the presence of pectin–chitosan gels can be a useful property of the serosal sealant.

The change in mechanical properties upon incubation in Hanks’ solution also depended on the type of cross-links in the gels studied. The hardness and elasticity of P–Ch0 and P–Ch1 gels, in the formation of which calcium cross-linking is involved, decreased at pH 5.0, 7.4, and 8.0. This was probably due to calcium leaching and the destruction of egg-box structures due to the formation of calcium hydrogen phosphate. Unlike calcium-cross-linked gels, the mechanical strength of chitosan-containing gels formed predominantly by links between -COO^−^ and -NH_3_^+^ groups increased upon incubation in Hanks’ solutions of pH 7.4 and 8.0. The improvement in mechanical properties was consistent with the displacement of water and the compaction of the gel network due to the decharging of the -NH_3_^+^ groups of chitosan. It can also be assumed that the divalent anions of the Hanks solution (e.g., SO_4_^2−^, H_2_PO^4−^) bind the positively charged amino groups of two adjacent chitosan chains, thereby forming additional cross-links. It is important that pectin–chitosan gels retain high values of elasticity and Young’s modulus. The dynamic movements of the intestinal wall, i.e., peristalsis, require the hydrogel adhesives to exhibit strong elasticity properties [5]. According to previous results [23], pre-incubation in Hank’s solution for 24 h significantly reduced the adhesion of the pectin gel to the serosa, while the adhesion of pectin–chitosan gels changed little. Therefore, our assumption was confirmed that chitosan-containing pectin gel could be more interesting as a serosal adhesive than chitosan-free pectin gel.

The potential mechanism of adhesion of pectin–chitosan gel to serosa was not elucidated in the present study. Chemically covalent contacts as well as physical interactions, including hydrogen bonds, ionic interactions, electrostatic interactions, and van der Waals forces, are among the intermolecular forces that hold adhesives and tissues together [3]. The low-adhesion serosal surface that covers the visceral organs is composed of mesopolysaccharides, whose structure remains poorly understood [56]. However, by analogy with mucoadhesion [57], the mechanism of adhesion to the serosa can be assumed to include at least two steps [20]. Initially, the contact area between the surfaces of the adhesive and the serosa increases during the wetting phase. Then the entanglement of the intercontact polymers happens [58]. The interpenetrated chains forming physical entanglements and weak chemical interactions resist detachment of the gel sample from the serosa during an adhesion test. Most likely, the polymer chains that are involved in gel network formation are hardly accessible for contact with the serosal substrate. Therefore, the galacturonan backbone is unlikely to provide adhesion to the serosa of pectin gel. The neutral side chains of pectin, including galactose, arabinose, and xylose residues, do not participate in gel cross-linking and, therefore, can interpenetrate with serosal mesopolysaccharides. Therefore, the low adhesive strength of the pectin gel may be due to the low content of neutral side chains (about 10%). In pectin–chitosan gels P–Ch1 and P–Ch2, not all polycationic chains of chitosan participate in the formation of the gel network and, therefore, are available for interaction with the negatively charged glycocalyx of the serous membrane. Given that the P–Ch1 and P–Ch2 gels’ surfaces had rougher microrelief than those of the P–Ch0 and P–Ch3 gels, it is possible that adhesiveness depends on the morphology of the gel surface. It can be assumed that the interaction of pectin and chitosan in our study formed the “globular scrambled-egg structurek”. Firstly, the chain lengths of pectin and chitosan used varied greatly (Mw 400 and 160–200 kDa, respectively). Secondly, the rigidity of both pectin and chitosan could be reduced in the presence of calcium ions since their adsorption reduces the repulsion of similarly charged groups along the polymer’s molecule. The low chain rigidity and difference in chain length of the reacting polyelectrolytes have been suggested to contribute to the dual structure conformation as “globular scrambled-egg structurek” [46]. The reason for the significant decrease in serous adhesion of the P–Ch3 gel remains unclear and may be due to poor mechanical properties.

Hydrogels for tissue regeneration of the serosa must allow the attachment and proliferation of fibroblasts. Fibroblasts are cells mainly responsible for collagen production; therefore, the strength of the anastomotic site depends on the presence of collagenous networks produced by fibroblasts. Previously, a number of studies have shown that pectin–chitosan biomaterials have a positive impact on the adhesion of mesenchymal stem cells [38,59], platelets [39], and osteoblasts [60,61]. The successful seeding of fibroblasts on the pectin–chitosan scaffold was shown by Michailidou [35]. In our study, an increase in the content of chitosan in pectin gel led to an increase in the number of adherent fibroblasts, which confirms previous data.

In conclusion, hybrid gels built by calcium cross-links of pectin molecules and cross-links of pectin and chitosan chains were prepared. Chitosan included in the pectin ionotropic gel was found to improve its mechanical and bioadhesive properties. The adhesion of pectin–chitosan gels with chitosan weight fractions of 0.25 and 0.50 to rat intestinal serosa ex vivo was significantly stronger than that of a chitosan-free pectin gel. The serosal adhesiveness of the pectin–chitosan gel significantly decreases with a further increase in the content of chitosan to weight fractions of 0.75. The increase in strength and number of linkages caused by chitosan provides good mechanical properties and high stability of the pectin–chitosan gel in physiological solutions with a wide pH range. Thus, serosal adhesion combined with high mechanical stability in a wide pH range appeared to be advantages of pectin–chitosan gel for the development of tissue adhesive to seal and mechanically support the intestinal anastomosis.

## 4. Materials and Methods

### 4.1. Materials

Apple pectin AU701 (galacturonic acid—89.5%, degree of methyl esterification—43%, Mw—401 kDa) (Herbstreith & Fox GmbH, Nuremberg, Germany) and chitosan (degree of deacetylation—>90%, Mw—160–200 kDa) (Tongxiang, Zhejiang, China) were used.

### 4.2. Preparation of Pectin–Chitosan Gels

Apple pectin (20 g) was dissolved in deionized water (500 mL). Chitosan (20 g) was dissolved in 0.4 M hydrochloric acid (500 mL). The solutions were heated to 90 °C under continuous magnetic stirring (200 rpm) for 60 min for better dissolution and then cooled to room temperature. Three different blend combinations of pectin and chitosan (3:1, 2:2, and 1:3) were made. Dialysis tubes with a pore size of 14 kPa (Sigma-Aldrich Co., St. Louis, MO, USA) were filled with the solutions obtained (80 mL) and held in a solution of 1.0 M calcium chloride (250 mL) for 48 h at 25 °C for gelling.

Cylinder-shaped (10–12 cm high) gels were formed. After being removed from the dialysis tubes, the gels were washed with distilled water and cut into single-serving cubes. Four types of gels were obtained: P–Ch0—4% AP; P–Ch1—3% AP mixed with 1% chitosan; P–Ch2—2% AP mixed with 2% chitosan; P–Ch3—1% AP mixed with 3% chitosan (Table 7).

The pH was determined for hydrogel aqueous homogenates (1:10 (*w*/*v*)) using an S20 SevenEasy™ pH meter (Mettler-Toledo AG, Schwerzenbach, Switzerland). The weight of 1 cm square hydrogel cubes (*n* = 8) was measured (AG245, Mettler Toledo International, Greifensee, Switzerland) to determine the density as weight/volume. Weight loss (WL) was determined using a gravimetric method [21]. The concentration of calcium was determined on homogenates of samples that were mechanically broken in a 1 M NaOH solution. The concentration of calcium was determined using a Calcium-Agat kit (Agat-Med, Moscow, Russia). The operation principle of the kit is based on the reaction of o-cresolphthalein complexone and calcium in an alkaline medium, resulting in the formation of a red-violet complex whose intensity is proportional to the concentration of calcium. The GalA content was determined by reaction with 3,5-dimethylphenol in the presence of concentrated sulfuric acid, as described earlier [25].

### 4.3. Scanning Electron Microscopy

A target-oriented approach was utilized for the optimization of the analytic measurements [62]. Before measurements, the cross-section of the sample was obtained using a scalpel at room temperature. The samples were mounted cut-up on a 25-mm aluminum specimen stub and fixed with carbon double-sided adhesive tape or colloidal graphite adhesive. Carbon coating with a thin film (25 nm) was performed. The observations were carried out using a Hitachi SU8000 field-emission scanning electron microscope (FE-SEM) (Hitachi High-Tech Corporation, Tokyo, Japan). Images were acquired in secondary electron mode at two accelerating voltages and at a working distance of 8–10 mm. The morphology of the samples was studied, taking into account the possible influence of a metal coating on the surface. EDS-SEM studies were carried out using the X-max 80 EDS system (Oxford Instruments, Abingdon, UK) at 30 kV accelerating voltage and a working distance of 15 mm.

### 4.4. Rheological Characterization

A rotational-type rheometer (Anton Paar, Physica MCR 302, Graz, Austria) equipped with a parallel plate geometry (diameter 25 mm; gap 4.0 mm) was used for the strain and frequency sweep measurements.

Strain sweep evaluation was performed from 0.01 to 100% strain amplitudes using a controlled shear rate mode at 20 °C at a constant frequency and stress of 1 Hz and 9.0 Pa, respectively. The storage modulus (G_’LVE_), loss modulus (G″_LVE_), loss tangent (tan δ) in LVE, complex modulus (G*_LVE_), limiting value of strain (the strain at which biopolymers enter from linear viscoelastic region to nonlinear viscoelastic region, γL), stress (τL), stress at flow point (τFP) with the corresponding complex modulus (G*FP), fracture stress (τFr), and slope of the loss tangent after flow point (tan δAF) were determined [49]. The shear strain dependence of G′ and G″ was determined using the power law equation:G′ = A′ × ω^n′^(3)
G″ = A″ × ω^n″^(4)
where ω is the angular shear strain (%); A′ (Pa × s) and A″ (Pa × s) are intercepts; and n′ and n″ are the slopes of G′ and G″ frequency dependence, respectively. A′ = A″ is a measure of the contribution of the viscous component in relation to the elastic component and represents the overall loss tangent of the material.

For the frequency sweep experiments, the obtained mechanical spectra were characterized by the values of G′ and G″ (Pa) as a function of frequency in the range of 0.3–70.0 Hz at 20 °C and a constant stress of 9.0 Pa. The loss factor tan δ was calculated as the ratio of G″ and G′ [63]. The power law function [64] was expressed as follows:η = K_c_ × y^n^(5)
where η is the steady viscosity, K_c_ is the consistency constant, y is the shear rate, and n is the power law index or flow behavior index.

The degrees of frequency dependence for G′ and G″ were determined by the power law parameters [65], which are expressed as follows:G′ = k′ × ω^n′^(6)
G″ = k″ × ω^n″^(7)
where G′ and G″ are the storage and loss moduli, respectively; ω is the oscillation frequency (Hz); and k′ and k″ are constants. In addition, the complex dynamic viscosity frequency dependence η × s was determined.

The strength of the network (A, Pa × s^1/z^) and the network extension parameter (z) were evaluated as follows, according to [66]:G′ = G* × ω = A × ω^1/z^(8)
where G* (Pa) is the complex modulus.

### 4.5. Texture Characterization

A puncture test (probe P/2, diameter 2 mm, depth 6 mm) for the P–Ch0, P–Ch1, P–Ch2, and P–Ch3 samples (1.0 cm high) was carried out using a TA-XT Plus (Texture Technologies Corp., Stable Micro Systems, Godalming, UK) instrument at room temperature.

The texture characterization of freshly prepared gels and gels after incubation in Hanks’ solution was determined. Gel cubes were incubated in 3 mL of Hanks’ solution (NaCl 140 mM, KCl 5 mM, CaCl_2_ 1 mM, MgSO_4_ 0.4 mM, MgCl_2_ 0.5 mM, Na_2_HPO_4_ 0.3 mM, KH_2_PO_4_ 0.4 mM, and D-glucose 6 mM) at pH 7.4, 5.0, and 8.0 for 24 h at 37 °C. Each Hanks’ solution variant was supplemented with 25 mM HEPES to maintain the set pH.

### 4.6. Tissue Adhesion Assay

The force of gel adhesion to the rat small intestine serosa was measured to evaluate the bioadhesive properties of the gels. Adhesion strength was measured using a texture analyzer, TA-XT Plus (Texture Technologies Corp., Stable Micro Systems, Godalming, UK). A gel probe was prepared by immersing a wooden toothpick in ethyl cyanoacrylate glue and piercing the gel cube with the glue end of the toothpick. Figure 15 shows a prepared gel probe. The probe compressed the rat small intestine serosa at 100 mN compression force for 20 s. The force of probe separation from the tissue after 20 s of pressing was recorded and calculated using Exponent Stable MicroSystems (Version V6.1.4.0, Godalming, UK).

To investigate the influence of the salt composition of the adhesion medium, the adhesion test was carried out on freshly prepared gels 3, 6, and 24 h after incubation in Hanks’ solution at pH 7.4, 5.0, and 8.0 supplemented with 10% fetal bovine serum. Each Hanks’ solution variant was supplemented with 25 mM HEPES to maintain the set pH. Gel probes for the adhesion test were prepared after incubation.

### 4.7. Statistical Analysis

All statistical analyses were performed using Statistica 10.0 (StatSoft, Inc., Tulsa, OK, USA). Results are presented as means ± standard deviations (SDs). The differences among the means in serum release, the rheological and mechanical parameters, and the digestion studies were estimated with one-way ANOVA and Tukey’s HSD test at *p* < 0.05.

## Figures and Tables

**Figure 1 marinedrugs-21-00375-f001:**
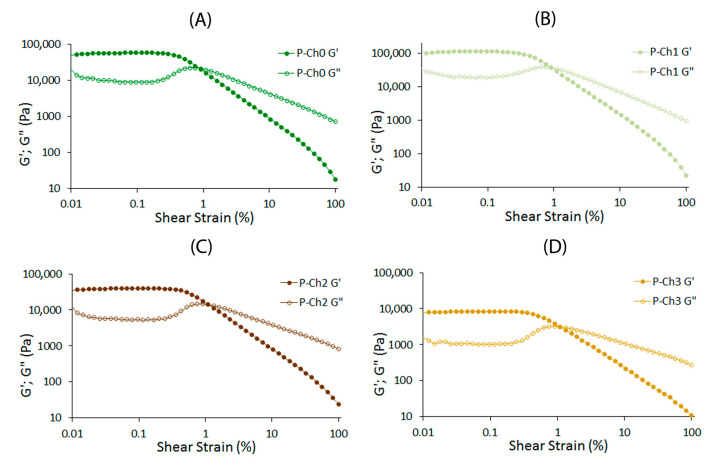
Strain sweep of P–Ch0 (**A**), P–Ch1 (**B**), P–Ch2 (**C**), and P–Ch3 (**D**) gels at a fixed frequency of 1 Hz and 20 °C.

**Figure 2 marinedrugs-21-00375-f002:**
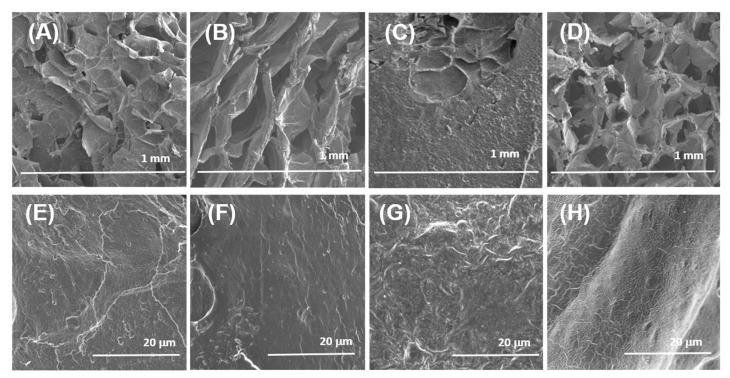
Scanning electron micrographs of the surfaces of the P–Ch0 (**A**,**E**), P–Ch1 (**B**,**F**), P–Ch2 (**C**,**G**), and P–Ch3 (**D**,**H**) gels. Magnification: (**A**–**D**)—50×, scale bar: 1.0 mm; (**E**–**H**)—2000×, scale bar: 20 μm.

**Figure 3 marinedrugs-21-00375-f003:**
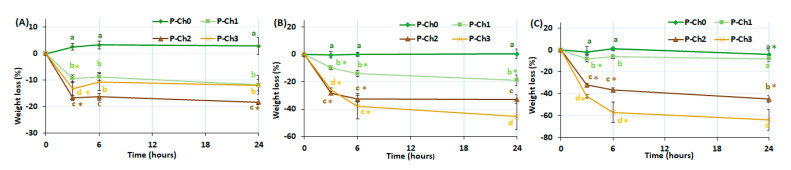
Weight change of pectin–chitosan gels during incubation in Hank’s solution of pH 5.0 (**A**), 7.4 (**B**), and 8.0 (**C**). Different lowercase letters a, b, c, and d indicate significant (*p* < 0.05) differences between gels. * *p* < 0.05 vs. previous time point. *n* = 6.

**Figure 4 marinedrugs-21-00375-f004:**
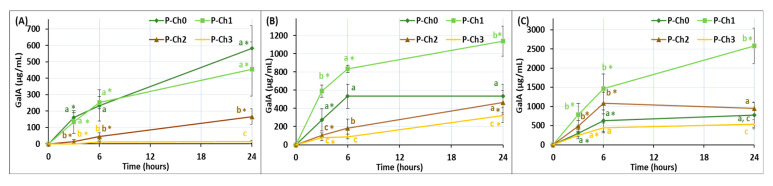
The content of GalA in the incubation medium during incubation of pectin–chitosan gels in Hank’s solutions of initial pH 5.0 (**A**), 7.4 (**B**), and 8.0 (**C**). Different lowercase letters a, b and c indicate significant (*p* < 0.05) differences between gels. * *p* < 0.05 vs. previous time point. *n* = 6.

**Figure 5 marinedrugs-21-00375-f005:**
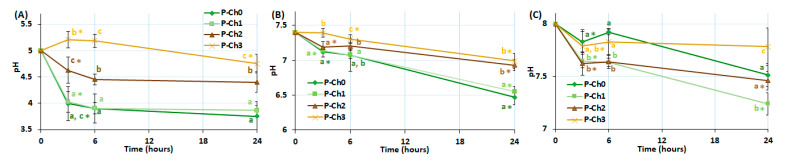
Change in pH level during incubation of pectin–chitosan gels in Hank’s solutions of initial pH 5.0 (**A**), 7.4 (**B**), and 8.0 (**C**). Different lowercase letters a, b and c indicate significant (*p* < 0.05) differences between gels. * *p* < 0.05 vs. previous time point. *n* = 6.

**Figure 6 marinedrugs-21-00375-f006:**
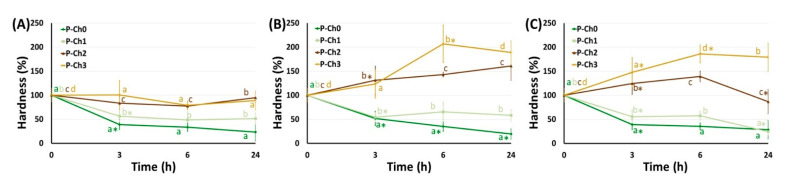
Change in hardness of pectin–chitosan gels during incubation in Hank’s solution of pH 5.0 (**A**), 7.4 (**B**), and 8.0 (**C**). Different lowercase letters a, b, c, and d indicate significant (*p* < 0.05) differences between gels. * *p* < 0.05 vs. previous time point. *n* = 8.

**Figure 7 marinedrugs-21-00375-f007:**
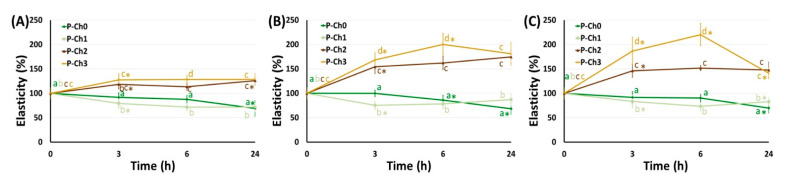
Change in elasticity of pectin–chitosan gels during incubation in Hank’s solution of pH 5.0 (**A**), 7.4 (**B**), and 8.0 (**C**). Different lowercase letters a, b, c, and d indicate significant (*p* < 0.05) differences between gels. * *p* < 0.05 vs. previous time point. *n* = 8.

**Figure 8 marinedrugs-21-00375-f008:**
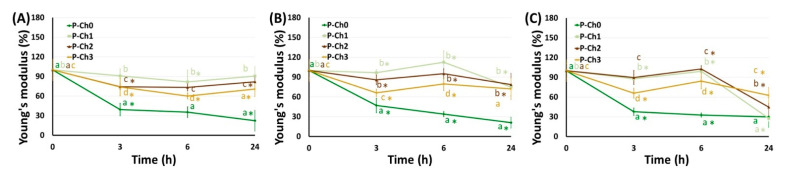
Change in Young’s modulus of pectin–chitosan gels during incubation in Hank’s solution of pH 5.0 (**A**), 7.4 (**B**), and 8.0 (**C**). Different lowercase letters a, b, c, and d indicate significant (*p* < 0.05) differences between gels. * *p* < 0.05 vs. previous time point. *n* = 8.

**Figure 9 marinedrugs-21-00375-f009:**
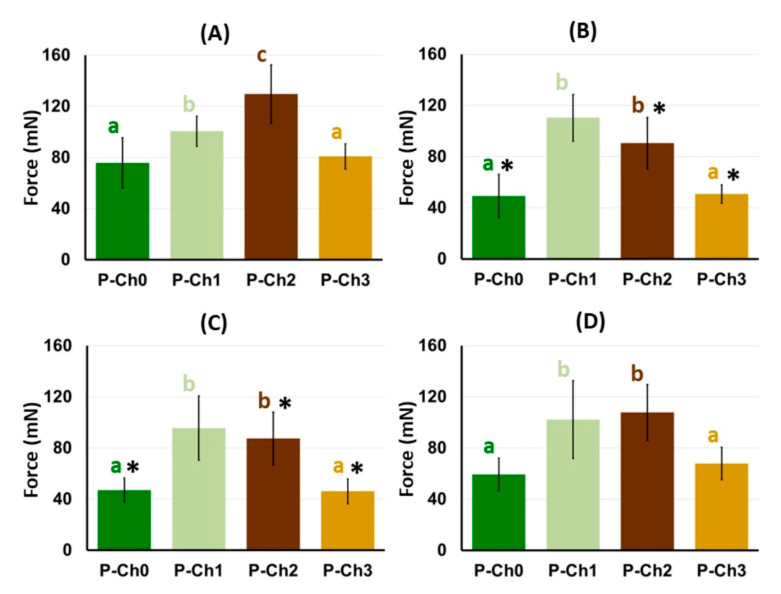
The adhesion strength of pectin–chitosan gels before (**A**) and after 24 h of incubation in Hank’s solution at pH 5.0 (**B**), 7.4 (**C**), and 8.0 (**D**). Different lowercase letters a, b and c indicate significant (*p* < 0.05) differences between gels. * *p* < 0.05 vs. before incubation. *n* = 8.

**Figure 10 marinedrugs-21-00375-f010:**
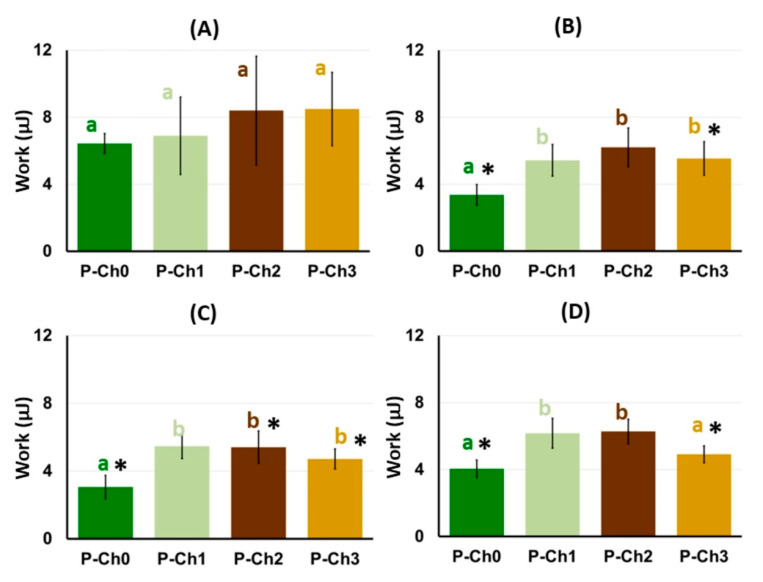
The work of adhesion of pectin–chitosan gels before (**A**) and after 24 h incubation in Hank’s solution at pH 5.0 (**B**), 7.4 (**C**), and 8.0 (**D**). Different lowercase letters a, b and c indicate significant (*p* < 0.05) differences between gels. * *p* < 0.05 vs. before incubation. *n* = 8.

**Figure 11 marinedrugs-21-00375-f011:**
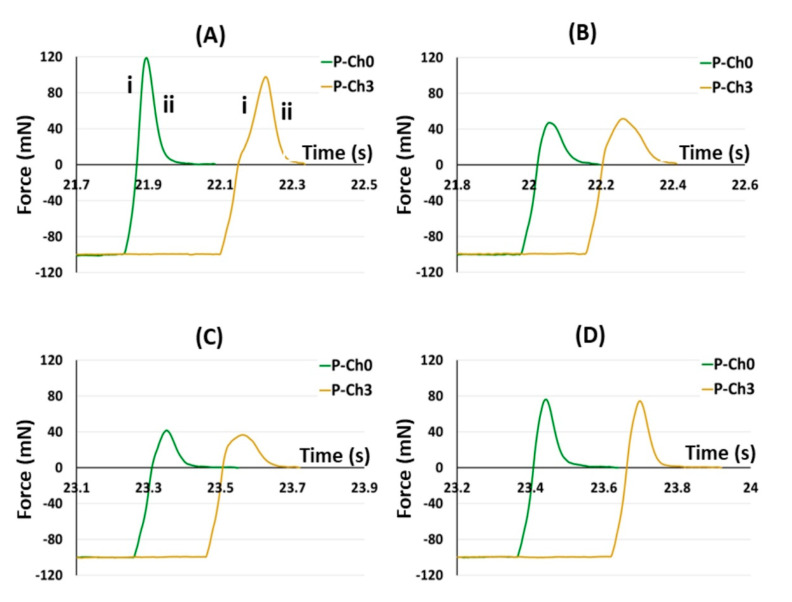
Representative tensile strength adhesive curve of the interaction of the P–Ch0 and P–Ch3 gels with the serosa before (**A**) and after 24 h incubation in Hank’s solution at pH 5.0 (**B**), 7.4 (**C**), and 8.0 (**D**). The deadhesion (i) and the debonding (ii) phases are shown.

**Figure 12 marinedrugs-21-00375-f012:**
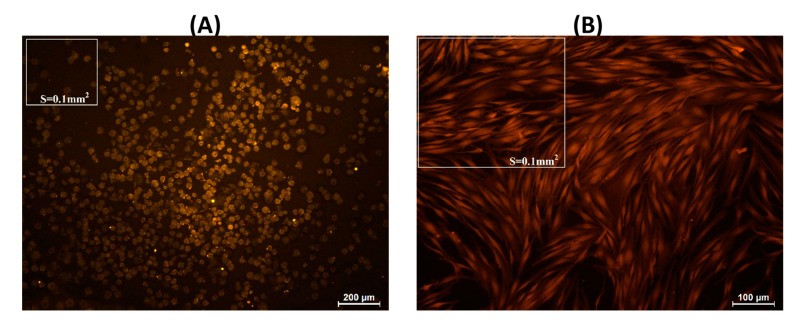
Adhesion of human fibroblasts to the plastic surface (control) after 2 (**A**) and 24 h (**B**) of incubation.

**Figure 13 marinedrugs-21-00375-f013:**
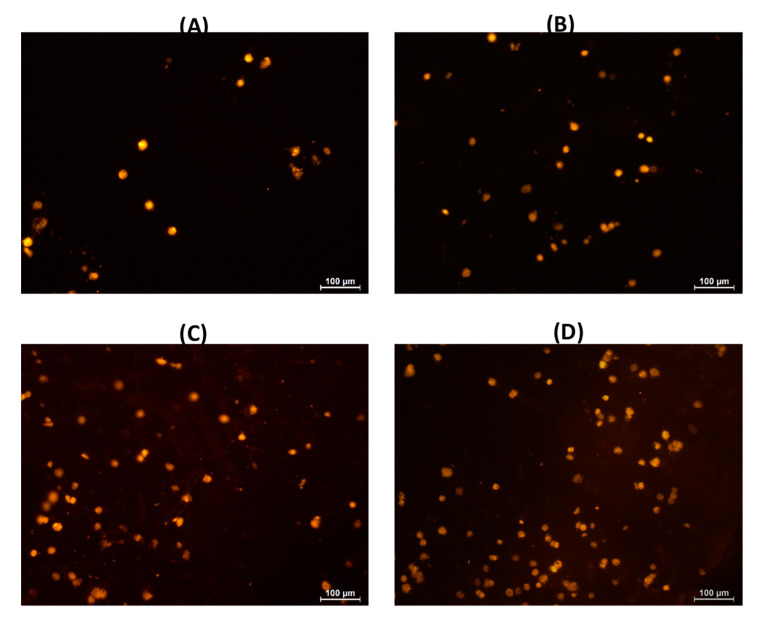
Adhesion of human fibroblasts to the P–CH0 (**A**), P–Ch1 (**B**), P–Ch2 (**C**), and P–Ch3 (**D**) gels after 24 h of incubation.

**Figure 14 marinedrugs-21-00375-f014:**
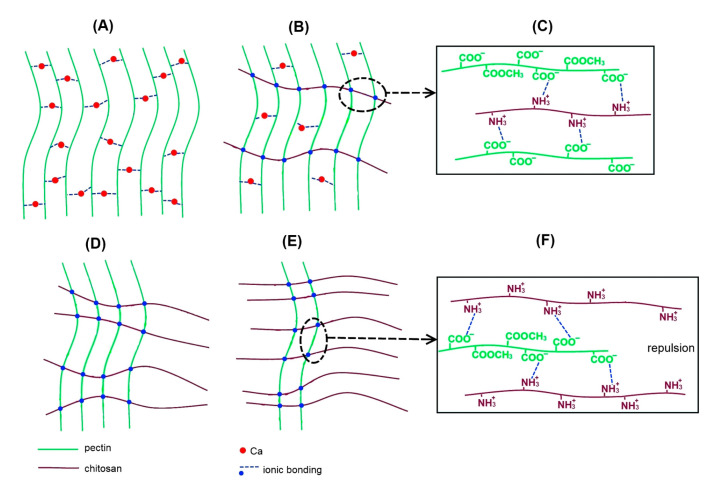
Mechanism presentation of the network model of the P–Ch0 (**A**), P–Ch1 (**B**), P–Ch2 (**D**), and P–Ch3 (**E**) gels. (**C**,**F**) show the possible interaction of pectin and chitosan chains in P–Ch1 and P–Ch3 gels, respectively.

**Figure 15 marinedrugs-21-00375-f015:**
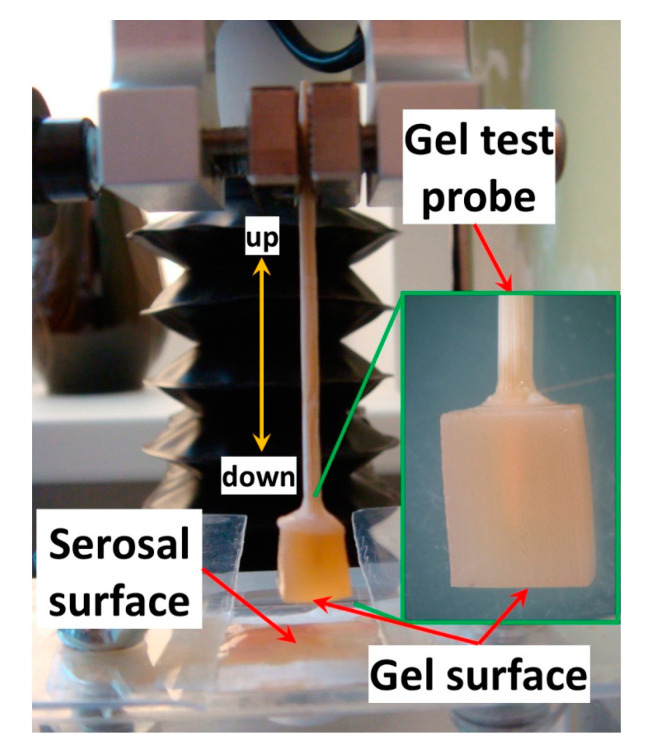
The gel probe for testing adhesion.

**Table 1 marinedrugs-21-00375-t001:** Effect of chitosan on the properties of pectin gel.

Gel Sample	Density (g/cm^3^)	Water Content (%)	pH	Hardness (N)	Elasticity (mm)	Young’s Modulus (kPa)
P–Ch0	1.04 ± 0.05 ^a^	94.3 ± 0.2 ^a^	3.99 ± 0.01 ^a^	1.11 ± 0.06 ^a^	1.5 ± 0.2 ^a^	2919 ± 324 ^a^
P–Ch1	1.03 ± 0.07 ^a^	94.0 ± 0.1 ^b^	3.54 ± 0.06 ^b^	3.27 ± 0.14 ^b^	2.6 ± 0.2 ^b^	4341 ± 488 ^b^
P–Ch2	1.05 ± 0.08 ^a^	95.2 ± 0.1 ^c^	3.03 ± 0.03 ^c^	1.68 ± 0.12 ^c^	1.8 ± 0.1 ^c^	3169 ± 420 ^a^
P–Ch3	1.04 ± 0.05 ^a^	97.4 ± 0.1 ^d^	3.47 ± 0.04 ^b^	0.42 ± 0.06 ^d^	1.9 ± 0.2 ^c^	781 ± 127 ^c^

Different lowercase letters ^a^, ^b^, ^c^, and ^d^ indicate significant (*p* < 0.05) differences between gels. *n* = 8.

**Table 2 marinedrugs-21-00375-t002:** Strain sweep for pectin–chitosan gels in the amplitude sweep test (1 Hz, 20 °C).

Parameters	P–Ch0	P–Ch1	P–Ch2	P–Ch3
G′_LVE_ (Pa)	56,286 ± 2324 ^a^	109,213 ± 5170 ^b^	38,438 ± 1365 ^c^	8159 ± 149 ^d^
G″_LVE_ (Pa)	10,660 ± 2647 ^a^	22,455 ± 4688 ^b^	6319 ± 1852 ^c^	1124 ± 138 ^d^
G*_LVE_	57,399 ± 1576 ^a^	111,755 ± 4130 ^b^	39,014 ± 991 ^c^	8240 ± 125 ^d^
Tan [δ]_LVE_	0.19 ± 0.06 ^a, c^	0.21 ± 0.05 ^a^	0.17 ± 0.06 ^ac^	0.14 ± 0.02 ^b, c^
γL (%)	0.36 ± 0.00 ^a^	0.31 ± 0.05 ^b^	0.42 ± 0.03 ^c^	0.39 ± 0.04 ^c^
τL (Pa)	50,774 ± 6348 ^a^	101,361 ± 29,599 ^b^	35,050 ± 3432 ^c^	7494 ± 995 ^d^
τFr (Pa)	21,900 ± 3359 ^a^	39,952 ± 10,169 ^b^	14,835 ± 1550 ^c^	3183 ± 296 ^d^
γFr	0.84 ± 0.14 ^a^	0.82 ± 0.19 ^a^	1.10 ± 0.00 ^b^	1.08 ± 0.15 ^b^
G*_FP_ (Pa)	31,661 ± 5293 ^a^	55,867 ± 14,392 ^b^	20,104 ± 2052 ^c^	4483 ± 443 ^d^
G*max/G*_LVE_	1.02 ± 0.01 ^a, c^	1.06 ± 0.06 ^a^	0.99 ± 0.03 ^a, c^	0.99 ± 0.02 ^b, c^
Tan [δ]_AF_	0.36 ± 0.06 ^a, b^	0.47 ± 0.10 ^a^	0.31 ± 0.04 ^a, b^	0.28 ± 0.04 ^a, b^

Different lowercase letters ^a^, ^b^, ^c^, and ^d^ indicate significant (*p* < 0.05) differences between gels. *n* = 6. Storage modulus (G′_LVE_), loss modulus (G″_LVE_), complex modulus (G*_LVE_), limiting value of strain (γL), limiting value of stress (τL), loss tangent (Tan [δ]_LVE_), fracture stress (τFr), corresponding complex modulus (G*_FP_) with the stress at flow point, and slope of the loss tangent after flow point (Tan [δ]_AF_), G*max/G*_LVE_ is the ratio of maximum complex modulus to linear complex modulus.

**Table 3 marinedrugs-21-00375-t003:** Frequency dependence of viscoelastic parameters for pectin–chitosan gels: viscosity and frequency (0.3 < ω < 70.0 or at 0.54/1.11/10.50/30.60/50.00 Hz).

Parameters			P–Ch0	P–Ch1	P–Ch2	P–Ch3
k′ (Pa × s)			56,226 ± 9345 ^a^	113,473 ± 10,527 ^b^	36,166 ± 2949 ^c^	7459 ± 470 ^d^
k″ (Pa × s)			7532 ± 1401 ^a^	15,675 ± 991 ^b^	4730 ± 280 ^c^	979 ± 49 ^d^
A			56,574 ± 9651 ^a^	113,440 ± 10,571 ^b^	35,560 ± 2809 ^c^	7497 ± 485 ^d^
k″/k′			7.50 ± 0.37 ^a^	7.23 ± 0.36 ^a^	7.64 ± 0.29 ^a^	7.62 ± 0.26 ^a^
η × s			8949 ± 1546 ^a^	17,966 ± 1636 ^b^	5690 ± 469 ^c^	1194 ± 83 ^d^
n′			0.10 ± 0.00 ^a^	0.10 ± 0.01 ^ab^	0.09 ± 0.01 ^b, c^	0.08 ± 0.02 ^c^
n″			0.05 ± 0.01 ^a^	0.04 ± 0.01 ^b^	0.01 ± 0.01 ^c^	0.01 ± 0.01 ^c^
z			9.5 ± 0.4 ^a^	9.5 ± 1.3 ^a^	10.2 ± 1.3 ^a^	13.6 ± 5.2 ^a^
Frequency (Hz)	0.54	G′ (kPa)	51,285 ± 8065 ^a^	102,928 ± 22,977 ^b^	32,858 ± 5877 ^c^	7035 ± 480 ^d^
		G″ (kPa)	10,678 ± 2588 ^a^	11,106 ± 2249 ^b^	3046 ± 660 ^c^	996 ± 43 ^d^
	1.11	G′ (kPa)	58,050 ± 9752 ^a^	115,778 ± 10,950 ^b^	37,197 ± 2761 ^c^	7494 ± 964 ^d^
		G″ (kPa)	8237 ± 1719 ^a^	17,779 ± 1534 ^b^	4569 ± 336 ^c^	491 ± 72 ^d^
	10.50	G′ (kPa)	72,130 ± 12,716 ^a^	145,137 ± 10,660 ^b^	44,835 ± 4858 ^c^	9034 ± 428 ^d^
		G″ (kPa)	8390 ± 1545 ^a^	16,897 ± 896 ^b^	3019 ± 253 ^c^	1044 ± 51 ^d^
	30.60	G′ (kPa)	79,186 ± 13,834 ^a^	157,555 ± 12,243 ^b^	47,215 ± 5114 ^c^	9416 ± 1157 ^d^
		G″ (kPa)	9105 ± 1806 ^a^	18,833 ± 1557 ^b^	3444 ± 369 ^c^	967 ± 61 ^d^
	50.00	G′ (kPa)	80,275 ± 14,949 ^a^	159,040 ± 17,820 ^b^	46,967 ± 5666 ^c^	8924 ± 3431 ^d^
		G″ (kPa)	10,435 ± 5166 ^a^	21,584 ± 11,068 ^b^	7550 ± 2878 ^c^	893 ± 446 ^d^

Different lowercase letters ^a^, ^b^, ^c^, and ^d^ indicate significant (*p* < 0.05) differences between gels. *n* = 6. The frequency dependences of the elastic (k′ and n′), loss (k″ and n″), and complex (A and z) moduli; the overall loss tangent (k′/k″); and the slope of complex viscosity (η × s).

**Table 4 marinedrugs-21-00375-t004:** Summary of power law parameters for the relationship between storage modulus or viscosity and frequency (0.03 < ω < 70.00 Hz or at 10 and 50 Hz) of pectin–chitosan gels.

Parameters	Viscosity				
	K_c_ (Pa × s)	*R* ^2^	*n*	η_app_ 10 (Hz)	η_app_ 50 (Hz)
P–Ch0	8951	0.999	−0.895	1099 ± 164 ^a^	263 ± 48 ^a^
P–Ch1	17,972	0.999	−0.896	2212 ± 161 ^b^	521 ± 59 ^b^
P–Ch2	5687	0.998	−0.913	682 ± 46 ^c^	154 ± 24 ^c^
P–Ch3	1190	0.998	−0.935	138 ± 7 ^d^	29 ± 11 ^d^

Different lowercase letters ^a^, ^b^, ^c^, and ^d^ indicate significant (*p* < 0.05) differences between gels. *n* = 6.

**Table 5 marinedrugs-21-00375-t005:** Mechanical properties of pectin–chitosan gels after 24 h of incubation at different pHs.

Gel Sample	Hardness (N)	Elasticity (mm)	Young’s Modulus (kPa)
pH 5.0			
P–Ch0	0.26 ± 0.21 ^a^	1.0 ± 0.2 ^a^	652 ± 316 ^a^
P–Ch1	1.69 ± 0.43 ^b^	1.9 ± 0.2 ^b^	3930 ± 643 ^b^
P–Ch2	1.59 ± 0.16 ^b^	2.3 ± 0.1 ^c^	2569 ± 211 ^c^
P–Ch3	0.37 ± 0.08 ^a^	2.4 ± 0.2 ^c^	552 ± 91 ^a^
pH 7.4			
P–Ch0	0.22 ± 0.12 ^a^	1.0 ± 0.2 ^a^	613 ± 239 ^a^
P–Ch1	1.91 ± 0.40 ^b^	2.3 ± 0.3 ^b^	3247 ± 705 ^b^
P–Ch2	2.70 ± 0.50 ^c^	3.2 ± 0.2 ^c^	2490 ± 539 ^c^
P–Ch3	0.79 ± 0.10 ^d^	3.4 ± 0.4 ^c^	562 ± 126 ^a^
pH 8.0			
P–Ch0	0.32 ± 0.21 ^a^	1.0 ± 0.2 ^a^	873 ± 460 ^a^
P–Ch1	0.80 ± 0.23 ^b^	2.2 ± 0.2 ^b^	1198 ± 203 ^a^
P–Ch2	1.46 ± 0.42 ^c^	2.7 ± 0.3 ^c^	1415 ± 465 ^b^
P–Ch3	0.75 ± 0.12 ^b^	2.6 ± 0.2 ^c^	488 ± 93 ^c^

Different lowercase letters ^a^, ^b^, ^c^, and ^d^ indicate significant (*p* < 0.05) differences between gels. *n* = 8.

**Table 6 marinedrugs-21-00375-t006:** Adhesion of human fibroblasts on the pectin–chitosan gels after 2 and 24 h of incubation.

Gel Sample	Incubation 2 h		Incubation 24 h	
	Adhesion (cells/0.1 mm^2^)	Cell Size (μm)	Adhesion (cells/0.1 mm^2^)	Cell Size (μm)
Control	52.4 ± 9.2 ^a^	27 ± 2 ^a^	59.1 ± 4.2 ^a^	101 ± 19 ^a,^*
P–Ch0	20.6 ± 5.6 ^b, c^	25 ± 2 ^a^	5.6 ± 0.9 ^b,^*	32 ± 8 ^b^
P–Ch1	24.2 ± 9.3 ^b^	24 ± 4 ^a^	12.1 ± 0.8 ^c,^*	24 ± 4 ^b^
P–Ch2	15.5 ± 2.1 ^c^	28 ± 2 ^a^	19.9 ± 2.3 ^d,^*	29 ± 5 ^b^
P–Ch3	18.0 ± 3.6 ^b, c^	26 ± 2 ^a^	25.9 ± 2.5 ^e,^*	23 ± 6 ^b^

Different lowercase letters ^a^, ^b^, ^c^, ^d^ and ^e^ indicate significant (*p* < 0.05) differences between gels. * *p* < 0.05 vs. 2 h incubation. *n* = 14.

**Table 7 marinedrugs-21-00375-t007:** Final compositions of the four pectin gel samples.

Gel	Apple Pectin (wt/v%)	Chitosan (wt/v%)
P-G0	4	-
P-G1	3	1
P-G2	2	2
P-G3	1	3

## Data Availability

The data that support the findings of this study are available from the corresponding authors upon reasonable request.

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
