# Peer review of "Effect of Chitosan on Rheological, Mechanical, and Adhesive Properties of Pectin–Calcium Gel"

_marinedrugs, 2023, doi:10.3390/md21070375_

Round 1
Reviewer 1 Report
Dear Author
You need to revise your results and figures carefully
- In introduction you have to provide the problem of rheological behavior of pectin and chitosan only and why you make this improvement (for which industrial field)
- Molecular weight of pectin and chitosan was missed … write them
- How did you confirm the presence of ca in your gel????
- In fig 2 (A), how the weight was decreased after 3 hour then increased by time increased at pH 5 and this behavior not shown at the other investigated pH
- There is no enough discussion for results
- In the discussion of Figure 3 is just listed the result , there is no reason even expected based on chemical or physical attraction, revise all your figure
- For the mechanical properties, I am very confused about the data after 6 h
o how the hardness sometimes decreased and sometime increased (give suitable reason)
o how the elasticity increased upon hardness decreased and reverse ??????????? (give suitable reason)
o in the elasticity and Young's modulus why ch1 showing different behavior than ch0, ch2, and ch 3
o
- In the preparation of your gel you wrote that you have put the reacted pectin and chitosan in solution of ca ion, and in the mechanism you suggest that the ca ion was make a cross linking between the pectin chain only. How you proved that????
- You wrote in Table 7. Final compositions of the five pectin gel samples, but it is only four composition you have work on it
- You wrote in viscosity result you have work at 10 and 50 Hz, but in the experimental part you wrote that you work at a constant frequency and stress of 1 Hz and 9.0 Pa, so, which is correct
Reviewer 2 Report
In their study, chitosan was incorporated into a pectin ionotropic gel to enhance its mechanical and bioadhesive properties. Pectin-chitosan gels, namely P-Ch0, P-Ch1, P-Ch2, and P-Ch3, with chitosan weight fractions of 0.00, 0.25, 0.50, and 0.75, were prepared and characterized through dynamic rheological tests, penetration tests, and serosal adhesion ex vivo assays. This is a very interesting finding.
I believe chitosan can also contribute to the seal-healing properties. Chitosan, a natural polymer obtained through the partial deacetylation of chitin in an alkaline solution, is primarily derived from the exoskeleton of crabs, shrimp, and krill. How about incorporating chitosan into their Pectin-Calcium Gel?
In the introduction, it would be beneficial to mention other types of gels, such as the Conical Frustum Gel driven by the Marangoni effect, which enables motor function without a stator, or the self-oscillating polymer gels that exhibit contraction waves, demonstrating the dynamic nature of gels.
The fascinating aspect of this study is that the authors conducted numerous experiments and assessed various characteristics of the gels. However, it would be helpful if they could provide some theoretical background as the performance of the gels is significantly influenced by the working environment.
In Figure 13, it is unclear how the authors demonstrated the Tissue Adhesion Assay, which might cause confusion. Clarification is needed in this regard.
It is noteworthy that their gels function within neural environments, offering promising applications in the field.
Round 2
Reviewer 2 Report
The authors diligently addressed my inquiries and incorporated abundant content into their manuscript, ensuring clarity for the readers. Thank you. I do not have any additional questions.